# Use of Bacteriophage Amended with CRISPR-Cas Systems to Combat Antimicrobial Resistance in the Bacterial Foodborne Pathogen *Listeria monocytogenes*

**DOI:** 10.3390/antibiotics10030308

**Published:** 2021-03-17

**Authors:** Cameron Parsons, Phillip Brown, Sophia Kathariou

**Affiliations:** 1Department of Food, Bioprocessing, and Nutrition Sciences, North Carolina State University, Raleigh, NC 27695, USA; skathar@ncsu.edu; 2Department of Plant and Microbial Biology, North Carolina State University, Raleigh, NC 27695, USA; pebrown4@ncsu.edu

**Keywords:** *L. monocytogenes*, antimicrobial resistance, CRISPR, listeriaphages

## Abstract

*Listeria monocytogenes* is a bacterial foodborne pathogen and the causative agent of the disease listeriosis, which though uncommon can result in severe symptoms such as meningitis, septicemia, stillbirths, and abortions and has a high case fatality rate. This pathogen can infect humans and other animals, resulting in massive health and economic impacts in the United States and globally. Listeriosis is treated with antimicrobials, typically a combination of a beta-lactam and an aminoglycoside, and *L. monocytogenes* has remained largely susceptible to the drugs of choice. However, there are several reports of antimicrobial resistance (AMR) in both *L. monocytogenes* and other *Listeria* species. Given the dire health outcomes associated with listeriosis, the prospect of antimicrobial-resistant *L. monocytogenes* is highly problematic for human and animal health. Developing effective tools for the control and elimination of *L. monocytogenes*, including strains with antimicrobial resistance, is of the utmost importance to prevent further dissemination of AMR in this pathogen. One tool that has shown great promise in combating antibiotic-resistant pathogens is the use of bacteriophages (phages), which are natural bacterial predators and horizontal gene transfer agents. Although native phages can be effective at killing antibiotic-resistant pathogens, limited host ranges and evolved resistance to phages can compromise their use in the efforts to mitigate the global AMR challenge. However, recent advances can allow the use of CRISPR-Cas (clustered regularly interspaced short palindromic repeats-CRISPR-associated proteins) to selectively target pathogens and their AMR determinants. Employment of CRISPR-Cas systems for phage amendment can overcome previous limitations in using phages as biocontrol and allow for the effective control of *L. monocytogenes* and its AMR determinants.

## 1. Introduction

*Listeria monocytogenes* is a facultative intracellular bacterial pathogen that is found ubiquitously in the environment and is most frequently transmitted via contaminated food [1,2,3]. *L. monocytogenes* is the causative agent of the disease listeriosis in humans and other animals. Of the 13 known serotypes of *L. monocytogenes*, three (i.e., 1/2a, 1/2b, and 4b) are primarily responsible for human listeriosis [4,5]. In susceptible individuals, listeriosis can result in severe symptoms including septicemia, meningitis, stillbirths, and abortions, and the case fatality rate remains high [5,6,7]. *L. monocytogenes* also exhibits an array of special environmental survival attributes, such as the capacity to grow at refrigeration temperatures, biofilm formation, and resistance to sanitizers and other antimicrobial compounds, which enable it to exhibit remarkable persistence in the environment and equipment of food-processing facilities [1,2]. Its severe disease potential coupled with its environmental survival characteristics render *L. monocytogenes* a major cause for public health concern.

A beta-lactam and an aminoglycoside [7,8]. *L. monocytogenes* exhibits innate resistance to nalidixic acid, oxacillin, and certain third–generation cephalosporins but is not commonly resistant to antibiotics employed for treatment [9,10,11]. However, there are several reports of resistance to clinically-relevant antibiotics in *L. monocytogenes* and other *Listeria* species [10,12,13,14]. Thus, antimicrobial resistance (AMR) in this serious human and animal pathogen is of concern, in agreement with the global trend of growing AMR challenges in agriculture and the food chain [15,16]. 

### 1.1. Bacteriophage as Biocontrol against Listeria monocytogenes and Tool to Mitigate AMR

Novel mitigation strategies are required in response to the growing threat that AMR poses to public health. Such strategies will be critically needed for *L. monocytogenes*, as it is one of the leading causes of death due to foodborne disease in the United States and other industrialized nations [6,17]. One potential solution to the problem of antibiotic-resistant *L. monocytogenes* involves the native predators and horizontal gene transfer agents of this pathogen, namely *Listeria*-specific bacteriophages (listeriaphages). These viruses have been extensively studied and approved for use against *L. monocytogenes* in foods and food-production facilities [18,19,20]. Listeriaphages have been shown to be effective at reducing *L. monocytogenes* populations in food matrices as well as in biofilms in food-processing plants and equipment [21,22,23,24]. While the potential for phages to control *L. monocytogenes* has been known for some time, they are still not widely employed. Some of the primary barriers to the more widespread adoption of phages for biocontrol include the limited host ranges of phages and the immunity of *L. monocytogenes* to phages [22]. With time, and repeated exposure, *L. monocytogenes* can develop resistance to phages, and strains from food-processing plants, especially those of serotypes 1/2a, 1/2b, and 1/2c, frequently exhibit resistance to phages [22,25,26,27,28].

One of the emerging tools in the fight against antibiotic-resistant bacteria that may also prove key to finally enabling the widespread use of listeriaphages for control of *L. monocytogenes,* including strains with specific AMR determinants, is the use of clustered regularly interspaced short palindromic repeats (CRISPRs). While CRISPR systems were first identified via their native function as a bacterial immune system [29], their targeted DNA manipulation capabilities have rendered them a leading tool for genome-editing purposes [30]. These DNA-encoded, RNA-mediated, DNA-targeting bacterial immune systems function by identifying invading exogenous DNA, and then targeting specific portions of the invading DNA for degradation by nucleases [31]. 

Given the intense interest in CRISPRs for genome editing purposes, and the massive increases in the number of sequenced bacterial genomes, the number and diversity of known CRISPR–Cas systems have dramatically increased. CRISPR systems are currently subdivided into two classes, six types, and 33 subtypes [32]. Class 1 systems have effector modules composed of multiple Cas proteins, whereas class 2 systems are characterized by a single multidomain protein that performs all functions necessary for interference (Figure 1). Class 1 systems have Cas6 for pre-CRISPR RNA (crRNA) processing into mature crRNA. Class 2 systems require trans-acting crRNA, which is usually transcribed within or near their locus and which functions in conjunction with their large multidomain proteins for crRNA maturation (Figure 1). In-vivo trials have already shown that these systems are capable of inactivating targeted pathogens in microbial communities as well as inactivating specific AMR genes [33,34]. Several studies have established the ability of CRISPR arrays to inactivate antibiotic-resistant microorganisms as well as specific AMR genes, in vivo [34,35,36,37,38,39,40]. In one such study, Kim et al. (2015) showed that CRISPR-mediated inactivation of a beta-lactam AMR gene in *Escherichia coli* rendered the bacteria susceptible to beta-lactam antibiotics [37]. Additionally, they found that for *E. coli* strains with multiple resistance genes harbored on the same plasmid as the beta-lactam resistance gene, double-stranded DNA breaks caused by the beta-lactam–targeting CRISPR resulted in the loss of the entire plasmid, which sensitized the bacteria to multiple antibiotics [37]. In another instance, the use of a CRISPR-equipped phage was found to be as efficacious as a high-dose fosfomycin administration in clearing a soft tissue *Staphylococcus aureus* infection [41]. If the use of CRISPRs can reduce the use of antibiotics, this would also be a means of decreasing the selection pressures for AMR. Such findings suggest that CRISPR systems can mitigate the AMR challenge since their employment can reduce the need for antibiotics and specifically inactivate AMR–conferring elements as well as the bacterial pathogens that harbor the corresponding elements.

CRISPR-Cas systems are widely disseminated and are encountered in roughly 40% of bacterial genomes [31]. Interestingly, while the use of endogenous CRISPR-Cas systems can yield superior results in genome editing [30,42], there have been relatively few studies utilizing native endogenous CRISPR systems. The available findings from these studies indicate that native systems are extremely effective for genome editing, and can alleviate the problems associated with exogenous systems, such as codon usage and the potential toxicity of exogenous Cas genes, while also being more efficient [30,42,43,44]. Appropriate CRISPR-Cas system selection and modification for inactivation of a target antibiotic-resistant pathogen is key for the success of utilizing CRISPR-Cas systems to mitigate AMR. 

### 1.2. CRISPR Systems in Listeria

Many earlier studies successfully detected CRISPR repeats in *L. monocytogenes* genomes [4,45,46,47]. These studies typically included relatively small numbers of genomes, making it difficult to ascertain the extent to which the findings might apply to *L. monocytogenes* as a species. A more recent survey of 128 *L. monocytogenes* genomes, comprised of 38 closed genomes from the National Center for Biotechnology Information (NCBI) and 90 genomes of isolates from food in North China, identified CRISPR arrays and corresponding Cas genes in 41.4% of the genomes. The CRISPR-Cas systems were exclusively type IB and IIA [48], in agreement with a previous work that primarily detected type IB or IIA systems in *L. monocytogenes* [49] (Figure 2). Type VI systems (Figure 2), though detected in *Listeria* species, have only been detected thus far in *L. seeligeri*, a species that is non-pathogenic to humans [50]. Several pieces of evidence suggest that at least some of the CRISPR-Cas systems detected in *L. monocytogenes* are functional. Specifically, many of the CRISPR arrays are sizable and with distinct spacer content from one another, and they harbor spacers that exactly match sequences of known *Listeria* phages and plasmids. Furthermore, they harbor complete sets of Cas genes without evidence for deleterious mutations [45,46,47,48,49,51]. Putatively-functional CRISPR-Cas systems have also been detected in other *Listeria* species that are not considered human pathogens, such as *L. seeligeri, L. ivanovii, L. innocua,* and *L. marthii* [47,50,51].

Interestingly, in *L. monocytogenes* the incidence of CRISPR-Cas systems appears to be serotype-dependent, as putatively functional CRISPR systems have been detected in all major serotypes except 4b [46,48]. Even though the underlying mechanisms remain to be elucidated, the finding is noteworthy, as serotype 4b appears to have enhanced virulence and includes several hypervirulent clones that have been repeatedly implicated in outbreaks [52,53].

The native presence of CRISPR-Cas systems in many *Listeria* genomes suggests their suitability for biotechnological applications against *L. monocytogenes,* though correct selection and adaptation of the systems will be crucial. There has been relatively little experimental work to determine the functionality of these in silico–identified systems in *L. monocytogenes.* One study demonstrated that the type II system in *L. monocytogenes* 10403S had weak activity against a plasmid harboring one of the spacers found in the native CRISPR array harbored by this strain [54]. Deletion of the Cas genes belonging to a type II system in *L. ivanovii* strain WSLC30167 resulted in the strain becoming highly sensitized to a phage to which it had previously been resistant [51]. Furthermore, mobilization of a self-targeting spacer into this strain via a plasmid resulted in massive cell death, indicating a functional and active CRISPR-Cas system [51]. Another study found the in silico–predicted type VI systems in *L. seeligeri* to be functional against invading conjugative plasmids, also demonstrating the functionality of these systems [50].

One major potential hurdle to the use of CRISPR-Cas systems for targeting *Listeria* is the widespread dissemination of anti-CRISPR proteins. These proteins have been shown to target and inactivate the Cas endonucleases of *Listeria* type II and type VI systems, and are highly prevalent on prophages that are widely distributed in *Listeria* [54,55]. Such findings suggest that the use of type II and type VI CRISPR systems, though widely used for such applications in other organisms, may be challenging in *Listeria.* The majority of experimental functional studies have involved type II systems and the associated nuclease Cas9, while more recently certain studies with other Gram-positive bacteria demonstrated that type I systems can be used as effectively as their more thoroughly-characterized type II counterparts [43,44,56]. Type I CRISPR systems are the most abundant in bacteria [57] and have also been extensively detected in *L. monocytogenes* [48]. In addition, they have been demonstrated to be effective in reversing AMR in *E. coli*, as discussed above [34]. However, these systems are seldom used for genome editing and comparatively little direct experimental work has been done to elucidate their complex function, particularly in *Listeria* [30]. Functional investigations of type I systems in *Listeria* are currently lacking. Such functional studies would afford a means of being able to employ type I systems as a potentially powerful tool in *Listeria.* It is worthy of note that no anti-CRISPR proteins targeting type I systems have been detected yet in *Listeria*, even though they have been identified in other bacteria [58,59,60].

### 1.3. Phage-Mediated Delivery of Engineered CRISPR-Cas Systems

While CRISPR-Cas systems have shown promise at combating AMR determinants and the host organisms in laboratory settings, they have yet to be extensively deployed in more complex settings. Real-world deployment of these systems shows significant promise given their extremely targeted nature. However, real-world deployment would require a delivery system for administering engineered CRISPR systems to antibiotic-resistant pathogens, including *L. monocytogenes.* Phages are an ideal delivery system as their natural function is to inject DNA into bacteria; they are extremely host-specific; they are generally considered safe for use in foods; and they have been shown to be stable in the environment [19,23]. CRISPR-equipped phages have been shown to be effective at inactivating human pathogens in vitro as well as in vivo [44,61]. AMR-targeting CRISPR systems have also been shown to effectively inactivate AMR determinants [35,36], and this is a highly promising means of addressing AMR in pathogens. While directly targeting the pathogen can potentially select for mutants capable of evading the CRISPR targets, targeting the resistance elements will render the pathogen antibiotic-sensitive, allowing the antibiotic to inactivate the pathogen before it can develop resistance to either the CRISPR system or the phage. The resulting inactivation of the targeted AMR determinants will minimize their potential to become disseminated via horizontal gene transfer. Furthermore, if the AMR gene targets are chromosomal, the resulting fragmentation by the Cas nuclease will also mediate the inactivation of the pathogen itself. Phages have been used to deliver a type 1 CRISPR-Cas system with synthetic spacers targeting AMR elements to *E. coli,* and were shown to be capable of killing the antibiotic-resistant bacteria as well as sensitizing resistant bacteria to antibiotics [34]. This perfectly demonstrates the efficacy of CRISPR-equipped phages for preventing the spread of AMR as well as for the targeted inactivation of antibiotic-resistant bacteria. 

Recent advances in molecular and synthetic biology have provided new tools for the modification of listeriaphages. Several studies have demonstrated the relative ease with which listeriaphages can be equipped with novel genetic content [62,63]. In one study, listeriaphages were genetically modified to encode an additional heterologous endolysin derived from an unrelated phage [62]. Upon lysis, the endolysins collectively attacked neighboring cells which included both phage-susceptible and phage-resistant cells, thereby circumventing phage resistance, currently a major impediment to using phages for biocontrol [62]. In another study, different variants of luciferase were incorporated into genomes of phages with different host ranges, allowing for the creation of fluorescent reporter phages that could quickly and easily differentiate among different types of *L. monocytogenes* in samples with mixed cultures [63].

Adding novel genetic content to phage genomes needs to be designed with caution to avoid exceeding the capacity of the phage head and creating impaired or non-functional phages, but correctly-designed modifications can create viable phages with novel functionalities. This is supported by the functional assessments of modified listeriaphages and other phages [34,62,63] and functional phage assessments combined with electron microscopy [44]. In addition to the ease with which listeriaphage genomes can now be modified, another study demonstrated that host ranges of listeriaphages can be expanded by modification of phage receptor binding proteins (RBPs) and the construction of chimeric phages harboring multiple RBPs on a single phage, allowing diverse host specificities [64]. Equipping a phage with multiple RBPs would enable CRISPR–equipped listeriaphages to target an even broader range of hosts. The incorporation of multiple chimeric phages into a single phage cocktail would further enhance the potential to combat target pathogens that have acquired resistance to specific phages.

## 2. Conclusions

While modified phages show great promise for the control of AMR and pathogens themselves, there are always concerns with the intentional release of any agent into an ecosystem. As previously mentioned, listeriaphages have been approved for use in foods since roughly 2006 [20]. Phage have been classified as generally recognized as safe (GRAS), and given that phages are the most abundant life form on the planet, humans and other animals are constantly exposed to them [65]. If their use is contained to clinical and food-processing environments, many of the concerns regarding their impact on the greater environment would also be reduced, as some CRISPR–equipped phages have already been used clinically without incident [41]. Additionally, phages are highly host-specific and therefore the overall impacts on the general microbial community would be reduced. While phages are a key driver of bacterial evolution [66], this role is primarily filled by temperate transducing phages. Lytic phages, particularly those equipped with host-targeting CRISPR systems, would likely kill their host at such a rate that they would not account for heritable changes in the bacterial genome. Phages with AMR-targeting CRISPR systems would indeed result in changes in the microbial genetic landscape, but those would primarily be the reduction in AMR genes in specific targeted pathogens, which are exactly the sort of changes needed to combat the global threat of AMR in pathogens. In conclusion, given the rising AMR threats globally, the dire health outcomes associated with listeriosis, and the critical roles of antimicrobial treatment in human and animal listeriosis, it is imperative to address AMR in *L. monocytogenes* before it becomes widespread in this pathogen. CRISPR-amended phages represent highly promising tools to address this issue. The employment of such engineered phages can inactivate AMR determinants in the targeted bacterial population before AMR can become widespread at the cost of human life, while also having the potential to inactivate the antibiotic-resistant pathogen itself. 

## Figures and Tables

**Figure 1 antibiotics-10-00308-f001:**
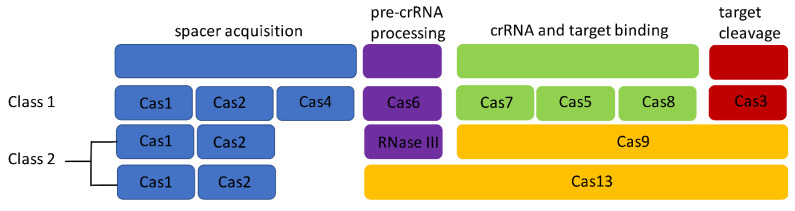
Composition of Class 1 and Class 2 clustered regularly interspaced short palindromic repeat (CRISPR) systems. Cas9 and Cas13 are the same color signifying they both perform multiple functions as indicated by their spanning multiple functional blocks.

**Figure 2 antibiotics-10-00308-f002:**
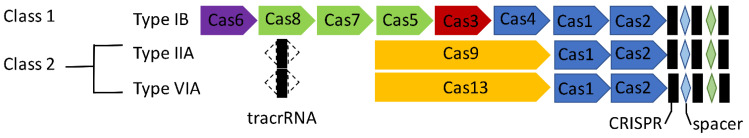
Composition of CRISPR types identified in *Listeria.* Colors here indicate distinct functional roles found in Figure 1.

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
