# Peer review of "Use of Bacteriophage Amended with CRISPR-Cas Systems to Combat Antimicrobial Resistance in the Bacterial Foodborne Pathogen Listeria monocytogenes"

_antibiotics, 2021, doi:10.3390/antibiotics10030308_

Round 1
Reviewer 1 Report
This a very nice paper with a high citation potential. However, the part dedicated to the use of CRISPR to fight AMR is not well developed in the manuscript. If this is because there are no many studies in the field, then, I would suggest changing the title and abstract in order to not mislead the reader about the focus of this paper.Author Response
RESPONSE: We would like to thank this reviewer for their time time and consideration of our work. We agree that adding more examples of work done specifically targeting CRISPR against AMR would help to develop one of the key points of this review. Additional references and aspects in regard to multidrug resistance have been added in the revision, please see lines 120-134.
Reviewer 2 Report
Please recheck the manuscript for spelling errors: eg. L. ivanovii instead of L. Invanovii; correct citation of references: [57] instead of (Meile et al 2020).
Suggestion: it would be nice if the authors would complete the manuscript with a more detailed description of the potential risks of using CRISPR equipped phages in food industry and animal feeding as well as the related environmental issues (eg. according to references 19 and 23).
Author Response
RESPONSE: We thank this reviewer for correctly identifying errors that we have corrected in our revised text. We have undertaken extensive error correction throughout. We agree that some of the greater ecological concerns regarding the use of CRISPR in this fashion would be important to discuss and have also included material to that effect in our updated manuscript, please see lines 259-278.
Reviewer 3 Report
The topic of work is very relevant by sanitary and veterynary point view but those envriomental too.
The work is balance tacking into account the clinical and animal but it should be improved integrating with environment information, such as environmental fate etc, in order to obtain fully ancl clearly of the intervention. In this way it can be better match the One Health approach.
Author Response
RESPONSE:We thank the reviewer for their time and evaluation of our work. We agree that this manuscript would benefit from a consideration of the broader ecological implications of using phage to control AMR, as such we have provided such a perspective in an expanded conclusions section please see lines 259-278.